



**Switchgrass ecotypes alter microbial contribution to deep**
**soil C**
**Damaris Roosendaal[1], Catherine E. Stewart[1,2], Karolien Denef[3], Ronald F. Follett[1*],**
**Elizabeth Pruessner[1], Louise H. Comas[4], Gary E. Varvel[5*], Aaron Saathoff[6], Nathan**
**Palmer[7], Gautam Sarath[7], Virginia L. Jin[5], Marty Schmer[5], Madhavan**
**Soundararajan[8]**
[1]{Soil-Plant-Nutrient Research Unit, United States Department of Agriculture-Agricultural
Research Service, Suite 100, 2150 Centre Avenue, Building D, Fort Collins, CO 80526-8119}
[2]{Natural Resource Ecology Laboratory, Colorado State University, Fort Collins, CO 80523-
1499, USA}
[3]{Central Instrument Facility (CIF), Department of Chemistry, Colorado State University, Fort
Collins, CO 80523-1872, USA}
[4]{Water Management Research Unit, United States Department of Agriculture-Agricultural
Research Service, Suite 100, 2150 Centre Avenue, Building D, Fort Collins, CO 80526-8119}
[5]{Agroecosystems Management Research Unit, USDA-ARS, 251 Filley Hall/Food Ind.
Complex, Univ. of Nebraska, Lincoln, NE 68583-0937}
[6]{LI-COR Biosciences, Lincoln, NE 68504, USA}
[7]{Grain, Forage, and Bioenergy Research Unit, USDA-ARS, 251 Filley Hall/Food Ind.
Complex, Univ. of Nebraska, Lincoln, NE 68583-0937}
[8]{Department of Biochemistry, University of Nebraska, Lincoln, NE 68588-0664}
[*]{Retired}
Correspondence to: catherine.stewart@colostate.edu



**Abstract**
Switchgrass (*Panicum virgatum* L.) is a $C_4$, perennial grass that is being developed as a
bioenergy crop for the United States.  While aboveground biomass production is well
documented for switchgrass ecotypes (lowland, upland), little is known about the impact of plant
belowground productivity on microbial communities down deep in the soil profiles. Differences
in root biomass and rooting characteristics of switchgrass ecotypes could lead to distinct
differences in belowground microbial biomass and microbial community composition. We
quantified root biomass and root architecture and the associated microbial abundance,
composition and rhizodeposit C uptake for two switchgrass cultivars using stable isotope probing
of microbial phospholipid fatty acids (PLFA) after $^{13}CO_2$ pulse-chase labeling. Kanlow, a
lowland cultivar with thicker roots, had greater plant biomass above- and belowground, greater
root mass density, and lower specific root length compared to Summer, an upland cultivar with
finer root architecture. The relative abundance of bacterial biomarkers dominated microbial
PLFA profiles for both Kanlow and Summer soils (55.4% and 53.5%, respectively), with
differences attributable to a greater relative abundance of gram-negative bacteria in Kanlow soils
(18.1%) compared to Summer soils (16.3%). The two ecotypes also had distinctly different
microbial communities process rhizodeposit C; greater relative atom % $^{13}C$ excess in gram-
negative bacteria (44.1 ± 2.3%) under the thicker roots of Kanlow and greater relative atom %
$^{13}C$ excess saprotrophic fungi under the thinner roots of Summer (48.5 ± 2.2%). For bioenergy
production systems, variation between switchgrass ecotypes could alter microbial communities
and impact C sequestration and storage as well as potentially other belowground processes.
**1   Introduction**
Switchgrass cultivars have been developed from ecotypes adapted to northern vs southern
latitudes and reflect trade-offs between plant productivity and stress resistance.  Upland ecotypes
are lower yielding with greater resistance to drought and freezing and lowland ecotypes are
higher yielding with poorer freeze tolerance traits (Fike et al., 2006; Garten et al., 2010; Monti,
2012).  Since switchgrass belowground biomass is proportional to or greater than aboveground
biomass in many switchgrass cultivars (Frank et al., 2004; Garten et al., 2010), greater
aboveground productivity in upland compared to lowland ecotypes may result in more root



biomass and thus more carbon (C) available as an energy substrate for belowground microbial
communities. Because most of the aboveground biomass is removed at harvest, the production
and dynamics of belowground biomass are important for potential soil C storage (De Deyn et al.,
2008; Garten et al., 2010). Very few switchgrass studies, however, examine if and how cultivar
influences soil microbial community abundance and composition by affecting rhizodeposit C,
particularly in deeper soil depths.

7         Surface soils are studied most intensely because the densities of soil microorganisms are

highest within organic matter and nutrient-rich surface soils (Fierer et al., 2003). Only limited
information is available for soil microbial communities deeper than 25 cm despite evidence that
more than half of the entire microbial community resides in subsurface soils (Van Gestel et al.,
1992; Dodds et al., 1996; Fritze et al., 2000; Blume et al., 2002). Because microorganisms are
involved in soil formation, ecosystem biogeochemistry, and groundwater quality (Fierer et al.,
2003), microbial dynamics in deeper soils are likely to exert considerable control on ecosystem
services, including C and nutrient cycles (De Deyn et al., 2008; Liang et al., 2012).

15        Soil C sequestration potential is determined by multiple factors such as topography,

mineralogy, and texture. Although microbial biomass represents a very small fraction of the total
soil C pool (Wardle, 1992), microbial metabolites stabilize soil organic carbon (SOC) and
provide plant nutrients, effectively driving plant C inputs into soils (De Deyn et al., 2008).
Intraspecific variability in switchgrass rooting architecture, structure, and root tissue could
produce differences in ecosystem C dynamics by affecting belowground C cycling and C
stabilization (de Graff et al., 2013) through both direct and indirect mechanisms on root
exudation and microbial community structure. While there is much uncertainty about the direct
impact of fine roots on soil C cycling, fine roots are one of the most important sources of soil C
input (Rasse et al., 2005; Joslin et al., 2006). Greater root exudation has been found in fast
growing plant species with branched, fine root systems (Personeni and Loiseau, 2004; De Deyn
et al., 2008). However, species with thicker roots may have a thicker cortical layer to support
more arbuscular mycorrhizal (AM) fungi (Brundrett, 2002; Comas et al., 2012; Comas et al.,
2014).   Previous switchgrass studies report that root architecture varies by cultivar or plant
genotype (Jackson, 1995; Fischer et al., 2006) and that upland switchgrass ecotypes have longer
specific root length (SRL) and finer root systems compared to coarser rooted lowland ecotypes



(de Graaff et al., 2013). What is less clear is if differences in root traits alter overall microbial
biomass and soil microbial community composition in the field.
Microbial phospholipid fatty acid (PLFA) analysis is a biochemical profiling technique to
evaluate soil microbial abundance and functional group composition (Vestal and White, 1989).
In addition, stable isotope probing of PLFAs following $^{13}CO_2$ pulse-labeling of plants can
determine which microbial groups are metabolizing recently produced rhizosphere-substrate
(Denef et al., 2007, Jin and Evans, 2010) as root exudates cycle through microbial biomass
quickly (de Graaff et al., 2014).
The objectives of this study were to determine the effect of differences in root traits
between two contrasting switchgrass cultivars on soil microbial biomass, soil microbial
community abundance and functional group composition, and microbial utilization of
rhizodeposit-C throughout the soil depth profile following $^{13}C$ pulse-labeling. We hypothesize
that the upland ecotype Summer will have finer roots, longer SRL, and greater specific surface
area, and that these traits will be associated with greater microbial biomass throughout the soil
profile compared to the lowland ecotype, Kanlow. We also hypothesize that rooting traits in
Kanlow will favor a greater relative abundance of soil fungi, particularly AMF, compared to
Summer due to lower specific root area.
**2   Materials and Methods**
**2.1 Experimental site and treatments**
The study site is located on the University of Nebraska-Lincoln's Agricultural Research
and Development Center (ARDC), Ithaca, Nebraska, USA (41.151°N, 96.401°W). Soils are
classified as Yutan silty clay loam (fine-silty, mixed, superactive, mesic Mollic Hapludalf) and
Tomek silt loam (fine, smectitic, mesic Pachic Argiudoll). The study is a randomized complete
experimental design with three field replicates of two switchgrass cultivars Summer and Kanlow.
Each plot consisted of twelve switchgrass plants of the same cultivar arranged in a 4 x 3 plant
grid. Switchgrass plants represent genetic individuals that were hand planted in summer 2009. At
the time of sampling for the current study, switchgrass was well-established and 3 years old.
Prior to the 2012 growing season, the plots were burned to remove aboveground biomass.





## 2.2 $^{13}$C labeling

All 12 switchgrass plants in each plot were labeled in May 2012 using a customized portable $^{13}CO_2$ pulse-chase labeling system consisting of a 1.0 m$^3$ clear polymethyl methacrylate (PMMA) chamber with an open bottom for placement over the entire plot and interior fans to provide air circulation (Saathoff et al., 2014). This chamber was attached to a Portable Photosynthesis System Model LI-6200 (Li-cor, Lincoln, NE) to monitor $CO_2$ concentration, air temperature and relative humidity within the chamber headspace. Isotopically enriched $CO_2$ label (99 atom% $^{13}$C (Sigma-Aldrich Co. St. Louis, MO)) was introduced into the chamber by opening the gas regulator for approximately 15 seconds. Label was added to raise chamber $CO_2$ concentrations between 1000 to 2000 ppm above atmospheric $CO_2$ concentration (420 ppm). Once the label was introduced, plants were allowed to take up labeled $CO_2$ until headspace concentrations were at least 100 ppm below ambient $CO_2$ levels.

## 2.3 Plant and soil sampling

Plants and soils for single, randomly selected individual switchgrass plants from each plot were harvested two days following $^{13}$C pulse-chase labeling. The aboveground biomass was removed by clipping at the soil surface. Plant samples were separated into tillers, stems, leaves, and oven dried at 55°C and ground for further analysis. Soil samples were then collected through the crown of the plant using a 10.16 cm diameter core attached to a hydraulic soil probe. Soil cores were divided in increments of 0-10, 10-30, 30-60, 60-90, 90-120, and 120-150 cm. Each depth increment was split in half length-wise, packed on ice, transported to the USDA-ARS laboratory in Ft. Collins Colorado, and refrigerated at 4°C until further processing. Soils were weighed, and a subsample was oven-dried at 110°C for 24 hours for determination of soil moisture content and soil bulk density. The half core for root separations was immediately frozen (-22°C). Samples for PLFA extraction and analysis were handpicked to remove all identifiable plant material, frozen at -22 °C and freeze-dried (Labconco FreeZone 77530, Kansas City, MO).

## 2.4 Root separations

The frozen half soil core was thawed to room temperature and the remaining plant crown was separated from roots and root samples were hand-washed. Specifically, roots were gently washed from the entire half core over a 1 mm (#20) soil sieve set over a second screen or sieve to capture all roots. Roots were picked off of the sieves and separated by hand into fine (1- 2



branches), 3$^{rd}$ order coarse roots, and coarse roots (4-5 order). Fresh root subsamples were
scanned with a desktop scanner to quantify morphological and architectural features (Comas and
Eissenstat, 2009). DT-SCAN software (Regent Instruments, Inc., Quebec, Canada) generated
length, average diameter, and volume of roots in each image, which were used to calculate root
length density (root length per soil volume, m cm$^{-3}$), specific root length (root length per root
mass, m g$^{-1}$), and root mass density (root mass per soil volume mg cm$^{-3}$). After scanning, root
samples were freeze-dried and then weighed. Root length and mass were scaled to the whole
core on a soil mass base using the weight of the ½ cores and the volume of the whole core.
Weight averages for the whole profile were scaled by depth increment using soil volume.
**2.5 Plant and soil analyses**
For the other half of the soil core, the crowns were separated from the roots, the soil was
sieved to 2 mm and all large roots and non-soil materials removed prior to soil characterization
and microbial analysis. Soil pH was determined with a Beckman PHI 45 pH meter using a 1:1
soil:water ratio. Total organic C, total N, and $\delta^{13}$C in both plant and soil samples were
determined in duplicate by a continuous flow Europa Scientific 20-20 Stable Isotope Analyzer
interfaced with Europa Scientific ANCA-NT system Solid/Liquid Preparation Module (Europa
Scientific, Crewe Cheshire, UK-Sercon Ltd.) Soil subsamples for PLFA analysis were
handpicked to remove all identifiable plant material, frozen at -22°C, then freeze-dried
(Labconco FreeZone 77530, Kansas City, MO) and stored at room temperature until lipid
extraction.
**2.6 PLFA extraction and quantification**
The extraction and derivatization of PLFAs was adapted from Bossio and Scow (1995)
and modified by Denef et al. (2007). Briefly, 6 g of soil from the surface depth increments (0-30
cm) and 8 g of soil from each subsoil depth increment (30-120 cm) were extracted using
phosphate buffer:chloroform:methanol in a 1:1:2 ratio. Total lipids were collected in the
chloroform phase, and fractionated on silica gel solid-phase extraction (SPE) columns
(Chromabond, Macherey-Nagel Inc., Bethlehem, PA) using chloroform, acetone, and methanol
as eluents. Polar lipid fractions representing PLFAs were collected from the methanol extractant
by mild alkaline transesterification using methanolic KOH to form fatty acid methyl esters
(FAMEs).



All PLFA samples were analyzed to identify and quantify individual PLFA biomarkers
using gas chromatography-mass spectrometry (GC-MS) (Shimadzu QP-20120SE) with a
SHRIX-5ms column (30 m length x 0.25 mm ID, 0.25 μm film thickness). The temperature
program started at 100 °C followed by a heating rate of 30 °C min$^{-1}$ to 160 °C, followed by a
final heating rate of 5 °C min$^{-1}$ to 280 °C. Prior to GC-MS analysis, a mixture of two internal
FAME standards (12:0 and 19:0) was added to the FAME extract. Individual fatty acids were
identified and quantified using these internal standards in addition to the relative response factors
for each of the external standard 37FAME and BAME mixes (Supelco Inc) as well as mass
spectral matching with the NIST 2011 mass spectral library.
The $\delta^{13}$C signature of individual FAMEs was measured by capillary gas chromatography-
combustion-isotope ratio mass spectrometry (GC-c-IRMS) (Trace GC Ultra, GC Isolink and
Delta V IRMS, Thermo Scientific). A capillary GC column type DB-5 was used for FAME
separation (30 m length x 0.25 mm ID x 0.25μm film thickness; Agilent). The temperature
program started at 60 °C with a 0.10 min hold, followed by a heating rate of 10 °C min$^{-1}$ to 150
°C with a 2 min hold, 3 °C min$^{-1}$ to 220 °C, 2 °C min$^{-1}$ to 255 °C, and 10 °C min$^{-1}$ to 280 °C with
a final hold of 1 min. The FAME $\delta^{13}$C values were calibrated using working standards (C12:0
and C19:0) calibrated on an elemental analyzer-IRMS (Carbo Eba NA 1500 coupled to a VG
Isochrom continuous flow IRMS, Isoprime Inc.). To obtain $\delta^{13}$C values of the PLFAs, measured
$\delta^{13}$C FAMEs values were corrected individually for the addition of the methyl group during
transesterification by simple mass balance (Denef et al., 2007; Jin and Evans, 2010).
Of the identified PLFAs, 2-OH 10:0, 2-OH 12:0, 2-OH 14:0, 16:1ω7, 17:0cy, 2-OH 16:0,
c18:1ω7, and 19:0cy are classified as gram-negative bacteria while  i-15:0, a-15:0, i-16:0, i-17:0,
and a-17:0 are classified as gram-positive bacteria, (Zelles, 1999). The 3-OH 12:0, 14:0, 15:0, 3-
OH 14:0, 17:0, and 18:0 are used as general bacterial indicators (Fröstegard and Bååth, 1996;
Zelles, 1999). The 16:0 fatty acid is classified as a universal PLFA (Zelles, 1999). The
10ME16:0, 10ME17:0 and 10ME18:0 are classified as actinomycete biomarkers. The 16:1ω5,
20:4ω6, 20:4ω3, and 20:1 are biomarkers for arbuscular mycorrhizal fungi (AMF) (Graham et.
al, 1995), and 18:3ω3, c18:2ω9,12, and c18:1ω9 are biomarkers for saprotrophic fungi (Zelles,
1997). Although 16:1ω5 can also be a gram-negative biomarker (Nichols, et al., 1986), in this





study the neutral lipid fatty acid (NLFA) fraction had high amounts of 16:1ω5, indicating
significant contribution from fungi (data not shown).

3       The abundance of individual PLFAs was calculated in absolute C amounts (ng PLFA-C

$g^{-1}$ dry soil) based on the PLFA-C concentrations in the liquid extracts, and used as a proxy for
microbial biomass. Changes in the microbial functional group composition were evaluated based
on shifts in PLFA relative abundances calculated and expressed as molar C percentage (mol%)
of each biomarker using the following formula:

$$\text{mol\%PLFA-C} = \frac{(\text{PLFA-C})_i}{\sum_{i=1}^{n}(\text{PLFA-C})_i} \times 100 \qquad (1)$$

where $(\text{PLFA-C})_i$ is the concentration of PLFA-C in solution (mol $L^{-1}$) and n is the total number
of identified biomarkers. Relative abundance values were then summed across all individual
biomarkers previously defined for each microbial functional group.

12       The ratio of fungi to bacteria was calculated as total fungal to total bacterial biomass

where total bacteria and fungi were determined by the sum of previously defined group
biomarkers as follows:
$\text{Bacteria}_{total} = \text{Gram-negative bacteria} + \text{Gram-positive bacteria} + \text{General bacteria}$
and
$\text{Fungi}_{total} = \text{AMF} + \text{Saprophytic fungi}$

18       Isotopic $^{13}$C enrichment in plant tissues and in soil microbial PLFAs were calculated as

atom percent enrichment (APE)
$\text{APE}\ ^{13}\text{C}_i = \text{atom\%}^{13}\text{C}_{labeled} - \text{atom\%}^{13}\text{C}_{unlabeled}$    (2)
for each i plant component (leaves, tillers, roots) or PLFA biomarker.
Label uptake by microbial functional group is then defined as:
$\text{APE}\ ^{13}\text{C}_{group} = \sum_{i=1}^{n} APE\ ^{13}\text{C}_i$    (3)
for $n$ functional group-specific biomarkers.
The relative distribution (%) of total label taken up that was recovered in each functional group
can then be calculated as:



1         Relative recovery$_{group}$ = APE $^{13}$C$_{group}$ / APE $^{13}$C$_{total}$ x 100,          (4)

where:

3         APE $^{13}$C$_{total}$ = $\sum_{i=1}^{m} APE\,^{13}C_i$              (5)

for $m$ total biomarkers identified, and other terms are previously defined.
Due to differing $^{13}$C label uptake between the two cultivars (Table 2), we express $^{13}$C enrichment
on a relative APE base (APE$_{rel}$ (Balasooriya et al. 2013)):

7         $\text{APE}_{rel} = \dfrac{APE\ 13Ci}{APE\ 13Ctotal} \times 100$           (6)

**2.7 Statistical Analyses**

9         A 2-way ANOVA with switchgrass cultivar and soil depth as main factors and plot as a

random effect was run for belowground plant biomass, soil %C, %N, bulk density, total PLFA-C
for each individual PLFA biomarker (ng PLFA C/g soil) and microbial group, and APE$_{rel}$ for
microbial groups using SAS v. 9.3 (SAS Institute, Cary, North Carolina, USA). Aboveground
biomass and plant biomass APE was run as a 1-way ANOVA with cultivar as the main effect
and plot as a random effect. Where necessary, data were log transformed to meet assumptions of
normality and equal variance. Treatments were considered significantly different for P ≤ 0.05
after Bonferroni adjustment.
**3   Results**
**3.1 Soil Properties**

20         Soil %C and %N decreased with soil depth (P < 0.0001) and pH increased with soil depth

(P = 0.003). For each depth increment, the soil characteristics beneath the two ecotypes were
generally similar (soil %C, %N, bulk density, pH and texture), except at the 120-150 cm depth
where %N was greater under Summer compared to Kanlow (P = 0.002, Table 1). There was no
significant effect of cultivar on bulk density (P = 0.9634, data not shown).
**3.2 Switchgrass Biomass**

26         The lowland cultivar Kanlow had more aboveground biomass (4886 ± 1220 g m$^{-2}$)

compared to Summer (1778 ± 660 g m$^{-2}$, P = 0.0153, Table 2). Total belowground root biomass



down to 150 cm was also greater in Kanlow (6633 ± 2165 g m$^{-2}$) compared to Summer (2271 ±
694 g m$^{-2}$, P = 0.029). This difference was driven by the top two depths (0-10 and 10- 30 cm),
which comprised 91% and 85% of root biomass for Kanlow and Summer, respectively.

**3.3 Root Characteristics**

Kanlow had significantly coarser, denser roots compared to Summer, resulting in a
shorter SRL throughout the soil profile, despite having similar root length densities (RLD)
(Table 3). Root mass density (RMD) was 2.8 to 6 times greater in Kanlow compared to Summer
in the first three soil depths and decreased with depth (Table 3). Weight averaged over the 0-150
cm profile, RMD was 5.48 ± 1.59 mg cm$^{-3}$ for Kanlow and 1.92 ± 0.69 mg cm$^{-3}$ for Summer (P =
0.001). However, the cultivars had similar root length densities (RLD) because the greater RMD
in Kanlow was comprised of roots with shorter SRL (Table 3). Kanlow's SRL averaged over the
soil profile was lower (25.96 ± 1.73 m g$^{-1}$ root) compared to Summer (52.66 ± 12.08 m g$^{-1}$ root,
P = 0.001).  The SRL for both ecotypes increased with depth as a result of lower RMD.

**3.4 Soil microbial biomass and community composition**

Differences in soil microbial biomass between ecotypes reflected differences in plant
productivity. The soils under Kanlow had greater PLFA-C (6.2 ± 0.2 µg PLFA-C g$^{-1}$ soil)
compared to Summer (4.7 ± 0.2 µg PLFA-C g$^{-1}$ soil) averaged across all depths (P = 0.0035,
Figure 1).  Total microbial biomass decreased with soil depth under both cultivars (P < 0.0001,
Figure 1) and the ecotype by depth interaction was also significant (P = 0.0019).  Total PLFA-C
decreased with depth under Summer, but increased in the 90-120 cm depth under Kanlow.
Despite the decreasing total PLFAs with depth, over half of the total observed PLFA biomass
was below 10 cm (Figure 1).
Soil microbial community composition differed between switchgrass ecotypes and
through the soil profile due to differences in bacteria (Figure 2).  Kanlow had relatively more
total bacterial PLFAs (55.4 vs. 53.5 % relative abundance, P = 0.0367), particularly more gram-
negative bacteria (18.1 % relative abundance) compared to Summer (16.3% relative abundance,
P = 0.0455) (Figure 2A). This resulted in the Kanlow soil microbial community having a
significantly lower gram-positive to gram-negative ratio (1.64) compared to Summer (1.88)
averaged over depths (P = 0.0165, Figure 3A).





In contrast, soils under Summer tended to have more fungal biomarkers and non-specific
microbial biomass biomarkers averaged over the soil profile compared to Kanlow soils (P =
0.140 and P = 0.0866, respectively). This resulted in marginally greater fungal:bacterial ratios
averaged over the profile (P = 0.064), particularly at the deeper depths (Figure 3B). There was
no difference between cultivars in microbial community structure in the 0-10 or 10-30 cm
depths.
A depth effect was observed in microbial community structure (P < 0.0001, Figure 2)
with gram-positive bacteria and actinomycetes being the most abundant in the 30-90 cm depths.
Actinomycetes increased to the 30-60 cm soil depth, then declined through the 150 cm depth
under both cultivars. Gram-positive bacteria followed a similar pattern, but peaked in the 60-90
cm depth increment before declining (P < 0.0001, Figure 2A). Bacteria increased with depth
initially, declined at the 30-60 cm depth, and then continued to increase through the 120-150 cm
depth (P < 0.0001, Figure 2A). Fungi and gram-negative bacteria were greatest at the surface and
deeper depths with a minimum at 30-60 cm or 60-90 cm depths (P < 0.0001, Figure 2A and 2B).

## 15  3.5 Plant $^{13}$C uptake

The $^{13}$C enrichment was detected in plant and root biomass throughout the soil profile 48
hours after labeling (Table 4). Enrichment was greater throughout the plant in Summer compared
to Kanlow with leaves 630 ± 113 vs. 474 ± 10 ng excess $^{13}$C g$^{-1}$ DM (P < 0.069) and tillers (1469
± 252 vs. 756 ± 110 ng excess $^{13}$C g$^{-1}$ DM, P < 0.007). Enrichment was also evident in labeled
roots throughout the soil profile and was generally greater in Summer vs. Kanlow and significant
in half the depths sampled (0-10, 10-30, 90-120 cm P < 0.0198). The root $^{13}$C enrichment was
similar within ecotype throughout the soil profile down to the 120-150 cm sample depth (Table

23  4).

## 24  3.6 $^{13}$C incorporation into microbial PLFAs

Microbial uptake of rhizodeposit C was observed in PLFAs throughout the profile to 150
cm after 48 hours. PLFA $^{13}$C enrichment for AMF, saprotrophic fungi, general bacteria, gram-
negative bacteria, gram-positive bacteria and universal microbial biomarkers was greater in the
pulse-labeled samples compared to the control (non-labeled) samples (Supplementary Tables 1
and 2). The two deepest depths (90-120 and 120-150 cm) should be interpreted with caution due
to large variation in the labeled PLFAs. Although total PLFA APE (ng excess $^{13}$C g$^{-1}$) was 1.78



times greater under Summer (10.97 ng excess $^{13}$C g$^{-1}$) compared to Kanlow (6.18 ng excess $^{13}$C
g$^{-1}$), it was not significant due to variability in individual plant and microbial $^{13}$C uptake (data not
shown). To normalize for these differences in $^{13}$C uptake, we express PLFA $^{13}$C enrichment as
relative atom % $^{13}$C excess (APE$_{rel}$) to compare between the two cultivars.

5       Relative rhizodeposit C uptake (APE$_{rel}$) under Kanlow was greatest in gram-negative

bacteria (44.1 ± 2.3% APE$_{rel}$, 16:1ω7, 17:0cy, 18:1ω7) and in saprotrophic fungi (50.6 ± 2.7%
APE$_{rel}$, c18:1ω9, 18:2ω9,12) under Summer (Figure 4) averaged over all depths. These
community differences became more pronounced through the soil profile, particularly in depths
deeper than 60 cm. Microbial communities in Kanlow soils had greater rhizodeposit uptake in
non-specific PLFAs (24.0 ± 1.7%, P= 0.006, 16:0) than Summer soils averaged over all soil
depths, and took up 32% of the rhizodeposited $^{13}$C label in the top two soil depths (P < 0.0001).
Rhizodeposit uptake in the AMF was dominant in biomarker 16:1ω5, did not differ between the
two cultivars, and decreased from 13.1 ± 1.3% relative enrichment in surface soils to 1.4 ± 2.4%
relative enrichment in the deepest soil layer (120-150 cm).
**4   Discussion**
**4.1 Ecotype root characteristics**
Switchgrass ecotypes have a broad range in phenology that reflects their adaptation
across a wide geographic area. The lowland ecotype, Kanlow, had 2.7 times more aboveground
and 2.9 times more belowground biomass than the upland cultivar, Summer. Although both
ecotypes allocated two-thirds of biomass belowground, there was a significant difference in
rooting traits throughout the soil profile. Differences between the two switchgrass ecotypes'
phenology were evident as the lowland ecotype, Kanlow, had significantly thicker roots with
shorter SRL compared to the upland cultivar, Summer.  The SRL for Summer (17.2 m g$^{-1}$ root
DW) was double that of Kanlow (8.3 m g$^{-1}$ root dry weight (DW)) in the 0-10 cm depth and
throughout the soil profile. DeGraaff et al. (2013) also found greater SRL in upland (253 ± 60
cm g$^{-1}$ DW) compared to lowland (170 ± 28 cm g$^{-1}$ DW) cultivars in the 0-15 cm depth across
eight switchgrass cultivars grown in IL.



Root mass density was two times greater under the lowland ecotype Kanlow than the
upland ecotype, Summer.  This is the opposite relationship found by Ma et al. (2000), who found
that the upland ecotype Cave-in-Rock had significantly greater RMD compared to the lowland
ecotypes Alamo and Kanlow in 7 year old switchgrass stands on a sandy loam in Alabama.
Variation between specific cultivars, soil nutrient status, soil texture, as well as climate
contributes to switchgrass rooting variability across sites and studies (Ma et al., 2000). Other
studies document cultivar-specific differences in root architecture between genotypes. Jackson
(1995) found root biomass cultivation and allocation were similar for lettuce (*Lactuca spp.*)
genotypes but their root architecture differed. Likewise, fine root morphology and architecture
are found to vary among species, apparently genetically determined and less plastic, while root
physiology appears to vary depending on current, whole plant metabolic activity (Comas et al.,
2004; Fischer et al., 2006).

**4.2 Effect of switchgrass cultivar on soil microbial community biomass and composition**

These differences in rooting characteristics resulted in different microbial biomass and
microbial community structure. In contrast to our hypothesis that Summer would have greater
microbial biomass, we found greater soil microbial biomass (PLFA-C) in Kanlow reflecting
greater belowground root biomass in Kanlow (Table 2 & Figure 1). The communities of the two
ecotypes also differed, with the lowland ecotype, Kanlow associated with a slightly more
bacterially-dominated soil microbial community than Summer. These community differences
could be a function either of microbial community modification by the plant from root exudation
(Broeckling et al., 2008; Gschwendtner et al. 2010) or root litter turnover and decomposition
(DeGraaff et al., 2013, 2014).  Plant cultivars have been shown to develop different microbial
rhizosphere communities (Broeckling et al., 2008; Gschwendtner et al. 2010) through root
exudation patterns (Broeckling et al., 2008). To our knowledge, this may be the first illustration
of switchgrass cultivar-specific impacts on soil communities in the field.
We observed greater fungal:bacterial ratios under the fine-rooted upland ecotype,
Summer, compared to the coarser rooted Kanlow over the profile, and the highest
fungal:bacterial ratio was found in the 120-150 cm depth. This was in contrast to our hypothesis
that Kanlow would have a more fungal community, particularly AMF. The finer rooting





architecture of Summer may promote greater root turnover and, in turn, promote a more
saprotrophic fungal community. It is interesting to note that there was no difference in the AMF
communities between the two cultivars, which may be a function of the thinner roots of Summer
having less cortex to support AM (Comas et al. 2014), or abundant N in this agronomic setting.
However, the presence of AM communities has been shown to stimulate root litter
decomposition, plant N uptake, and saprotrophic fungal abundance without altering AM
abundance (Herman et al. 2012).
**4.3 Effect of depth on soil microbial community abundance and composition**
There was an overall decrease in the total microbial biomass ($\mu$g PLFA-C g$^{-1}$ soil) with
depth (Figure 1) which corresponds to previous studies (Fierer et al., 2003; Aliasgharzad et al.,
2010; Kramer and Gleixner, 2008). Because soil microbes primarily use C from root exudates as
their energy source and C availability decreases with soil depth (Table 2), microbial biomass is
also expected to decline (Chaudhary et al., 2012).
Microbial community structure also changed with depth. Our results for 0-60 cm soils
agree with those of Fierer et al. (2003), who found gram-positive bacteria and actinomycetes
increased in proportional abundance with increasing soil depth and that gram-negative bacteria
and fungi were highest in surface soils. In the current study, the proportion of total PLFAs
attributable to fungi (saprotrophic fungi and AMF) was generally higher in surface soils than
deeper soils and that fungi and gram-negative biomarkers decreased with depth (0-60 cm). More
specifically, fungi and gram-negative PLFAs decreased in proportional abundance down through
60 to 90 cm in depth and subsequently increased through the 120 cm depth profile while gram-
positive and actinomycetes PLFAs showed the opposite trend, increasing in proportional
abundance through 60 to 90 cm in depth and decreasing through the remainder of the 120 cm
depth profile.
Previous studies have shown that higher available C or rates of C addition to soil tend to
have greater proportional abundance of fungi and gram-negative bacteria while gram-positive
and actinomycetes are proportionately lower under the same conditions (Griffiths et al., 1999;
Fierer et al., 2003). Thus in depths that are C-rich we should expect higher proportions of fungi
and gram-negative bacteria and in areas of C limitation we should expect higher proportions of
gram-positive and actinomycetes. This suggests more microbial C-limitation at the middle of the



depth profile, perhaps reflecting the high soil C content near the surface and active plant root exudation deeper in the profile.

**4.4 Microbial rhizodeposit-C utilization**

Microbial uptake of rhizodeposit $^{13}$C was observed in PLFAs throughout the soil profile to 150 cm depth 48 hrs post-labeling and illustrated distinct microbial community uptake patterns between switchgrass ecotypes, particularly deeper than 60 cm. The majority of labeled rhizodeposit uptake under Kanlow was by gram-negative bacteria which took up 44.1 ± 2.3% of the total $^{13}$C label recovered from all biomarkers whereas under Summer the rhizodeposit uptake was predominantly by the saprotrophic fungi (48.5 ± 2.2% relative enrichment) (Figure 4). These microbial community differences could be a function either of microbial community modification by the plant from root exudation (Broeckling et al., 2008; Gschwendtner et al. 2010) or root litter turnover and decomposition (DeGraaff et al., 2013, 2014). Although we did not measure root exudation here, other studies have documented that cultivar differences in root exudation influence microbial community structure (Gschwendtner et al., 2010; Marschner et al., 2001).

The differing rhizodeposit uptake patterns in the microbial communities associated with the two cultivars illustrated differing active plant-microbial associations. Kanlow, with thicker roots, may have greater root exudation and promote more endophytic bacterial associations. Gram negative bacterial endophytes (Protobacteria) are associated with switchgrass and have been shown to increase switchgrass growth (Xia et al., 2012). The finer root system of Summer may have exudation patterns that promote decomposition by saprotrophic fungi as a means for recovering nutrients from fine-root turnover. Recent work suggests that plants may promote litter decomposition for nutrient acquisition (Herman et al., 2012).

Although rhizodeposit uptake by AMF biomarkers did not differ between the two switchgrass cultivars and only comprised 13% of total enrichment in the 0-10 cm soil depth, soil fungi have the potential to strongly affect soil C sequestration. Fungal mycelia are comprised of complex, nutrient-poor carbon forms like chitin and melanin, allowing fungal metabolites to reside longer in soil than bacteria whose membranes mainly consist of phospholipids that are quickly reincorporated by soil biota (Rilling and Mummey, 2006; Six et al., 2006; De Deyn et al., 2008). By immobilizing C in their mycelium, extending root lifespan, and improving C




sequestration in soil aggregates mycorrhizal fungi can reduce soil C loss (Langley et al., 2006;
Rillig and Mummey, 2006; De Deyn et al., 2008).
**4.5 Impacts for bioenergy production & C sequestration**

5       Switchgrass is a strong candidate for soil C sequestration due to its fibrous root system

that can extend through a depth of 3 m (Ma et al., 2000; Liebig et al., 2005; Schmer et al., 2011).
Previous studies have shown that switchgrass has the capacity to increase SOC, mitigate
greenhouse gas emissions, and improve soil quality (Sanderson et al., 1999; Garten et al., 2000;
Frank et al., 2004; Liebig et al., 2005; Stewart et al., 2014). Furthermore, results from previous
studies indicate that switchgrass is effective at storing SOC below depths of 30 cm, not just near
the soil surface (Sanderson et al., 1999; Garten et al., 2000; Follett et al. 2012; Liebig et al.,

12  2005).

Garten et al. (2010) studied differences in above and belowground biomass in addition to
soil C stocks and N stocks for varying 3 year-old switchgrass plant cultivars. They found no
significant difference among lowland cultivars for total aboveground or belowground biomass, C
stocks, or N stocks in the 0-90 cm soils sampled in their study. In contrast to their observations,
our results indicate cultivar differences in root production and soil microbial communities in
only 3 year-old switchgrass plants through a soil depth profile of 150 cm for the two cultivars
Kanlow and Summer. It should be noted that the cultivars within the study done by Garten et al.
(2010) contained only lowland ecotypes whereas our study is comparing a lowland ecotype
(Kanlow) to an upland ecotype (Summer). Our results suggest Kanlow as higher yielding for
aboveground biomass, belowground root biomass and promoting total soil microbial biomass
(Table 2, Figure 1), but Summer may have a greater potential for soil C sequestration due to
greater C transfer to the soil fungal community and promotion of soil aggregation.
**5   Conclusions**
The two switchgrass ecotypes had distinct differences in root biomass and morphology
that resulted in differences in the associated soil microbial biomass, microbial community
composition and rhizodeposit C uptake. The lowland ecotype had significantly greater RMD but





similar RLD due to having shorter SRL compared to the upland ecotype, Summer. Kanlow had
more microbial biomass and a more bacterial dominated microbial community than Summer.
Although the differences between cultivar microbial communities was modest, rhizodeposit
uptake was quite different between ecotypes. The rhizodeposit C was processed primarily by
gram negative bacteria under Kanlow and saprotrophic fungi under Summer. Variation in
microbial community composition as well as rhizodeposit C uptake could result in different C
sequestration dynamics. For bioenergy production systems, variation between switchgrass
ecotypes could impact C sequestration and storage as well as potentially other belowground
processes by altering microbial communities and their role in C processing.
**Acknowledgements**
The authors acknowledge field assistance technician Nathan Mellor and our support scientist
Paul Koerner and laboratory assistance from Tamara Higgs, Erin Grogan Amber Brandt, Jordan
Wieger, Philip Korejwo and numerous student workers. This work is part of the USDA-ARS
GRACEnet Project: http://www.ars.usda.gov/research/GRACEnet.  This work was supported
in part by the Office of Science (BER), U. S. Department of Energy Grant Number DE-AI02-
09ER64829, and by the USDA-ARS CRIS project 5440-21000-030-00D.   The U.S.
Department of Agriculture, Agricultural Research Service, is an equal
opportunity/affirmative action employer and all agency services are available without
discrimination.  Mention of commercial products and organizations in this manuscript is
solely to provide specific information. It does not constitute endorsement by USDA-ARS
over other products and organizations not mentioned.



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



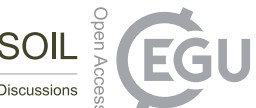

1 Table 1. Soil properties (%C, %N, texture, pH) for switchgrass lowland (cv. Kanlow) ecotype and upland ecotype (cv. Summer) down
2 to 150 cm. Values in parentheses are standard deviations.

| Cultivar | Soil Depth (cm) | SOC (%) | Total N (%) | Texture[†] | pH |
|---|---|---|---|---|---|
| Kanlow | 0-10 | 2.29 (0.05) | 0.20 (0.00) | silty clay loam | 6.24 (0.21) |
| | 10-30 | 1.62 (0.05) | 0.14 (0.00) | silty clay loam | 6.32 (0.24) |
| | 30-60 | 1.26 (0.05) | 0.11 (0.00) | silty clay loam | 6.48 (0.15) |
| | 60-90 | 0.57 (0.05) | 0.05 (0.00) | silty clay loam | 6.60 (0.12) |
| | 90-120 | 0.34 (0.06) | 0.04 (0.01) | silty clay loam/silt loam | 6.66 (0.15) |
| | 120-150 | 0.22 (0.07) | 0.03 (0.01) | silt loam | 6.90 (0.12) |
| | | | | | |
| Summer | 0-10 | 2.11 (0.05) | 0.18 (0.00) | silty clay loam | 5.92 (0.60) |
| | 10-30 | 1.60 (0.05) | 0.14 (0.00) | silty clay loam | 6.19 (0.57) |
| | 30-60 | 1.12 (0.05) | 0.10 (0.00) | silty clay loam | 6.64 (0.29) |
| | 60-90 | 0.56 (0.05) | 0.06 (0.00) | silty clay loam | 6.61 (0.19) |
| | 90-120 | 0.34 (0.05) | 0.04 (0.01) | silty clay loam/silt loam | 6.70 (0.19) |
| | 120-150 | 0.25 (0.01) | 0.04 (0.01) | silt loam | 6.83 (0.01) |

4 [†] from NRCS (https://soilseries.sc.egov.usda.gov/OSD_Docs/Y/YUTAN.html)





Table 2. Aboveground plant biomass (including crowns) and belowground root biomass per ground area (g m$^{-2}$) and standard
deviation (in parenthesis) for switchgrass lowland (cv. Kanlow) ecotype and upland ecotype (cv. Summer). P-values equal to or below
0.05 indicates whether the difference in biomass is significantly different between Kanlow and Summer in the aboveground plant
sampling, the total root biomass, and at every individual sampling depth.

| | Kanlow | Summer | P-value |
|---|---|---|---|
| | | (g m$^{-2}$) | |
| Aboveground Biomass | 4886 (1220) | 1778 (660) | 0.0153 |
| | | | |
| Root Biomass by Depth | | | |
| 0-10 cm | 4212 (1193) | 1652 (712) | 0.009 |
| 10-30 cm | 1826 (1059) | 272 (108) | <0.0001 |
| 30-60 cm | 253 (52) | 134 (43) | 0.068 |
| 60-90 cm | 110 (14) | 105 (45) | 0.775 |
| 90-120 cm | 105 (51) | 78 (43) | 0.422 |
| 120-150 cm | 126 (23) | 57 (17) | 0.044 |
| Total Root Biomass | 6633 (2165) | 2271 (694) | 0.029 |



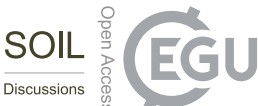

Table 3. Root mass density (mg cm$^{-3}$) root length density (cm cm$^{-3}$ soil), and specific root length (m g$^{-1}$ root) and standard deviation in
parenthesis for switchgrass lowland ecotype (cv. Kanlow) and upland ecotype (cv. Summer).

| Depth | *Root Mass Density* | | | *Root Length Density* | | | *Specific root length* | |
|---|---|---|---|---|---|---|---|---|
| | **Kanlow** | **Summer** | | **Kanlow** | **Summer** | | **Kanlow** | **Summer** |
| **(cm)** | **(mg cm$^{-3}$)** | | | **(cm cm$^{-3}$ )** | | | **(m g$^{-1}$ root)** | |
| 0-10 | 21.65 (5.30) | 8.26 (3.56) | *** | 18.00 (4.23) | 13.63 (4.02) | | 8.33 (0.09) | 17.22 (2.63)** |
| 10-30 | 4.89 (2.84) | 0.76 (0.34) | *** | 5.54 (0.17) | 2.77 (0.17)* | | 15.71 (9.26) | 39.64 (13.54)*** |
| 30-60 | 0.46 (0.17) | 0.24 (0.08) | * | 0.97 (0.35) | 1.11 (0.15) | | 21.42 (6.30) | 48.40 (8.85)*** |
| 60-90 | 0.19 (0.02) | 0.17 (0.06) | | 0.54 (0.04) | 1.46 (0.51)*** | | 31.49 (5.16) | 88.12 (1.59)*** |
| 90-120 | 0.19 (0.09) | 0.18 (0.09) | | 0.93 (0.14) | 0.99 (0.21) | | 52.85 (16.00) | 69.91 (46.17)*** |
| 120-150 | 0.22 (0.02) | 0.11 (0.03) | | 1.18 (0.35) | 1.43 (0.76) | | 60.83 (13.85) | 128.63 (34.72)*** |
| | | | | | | | | |
| *0-150* | *5.48 (1.59)* | *1.92 (0.69)* | *** | *5.20 (1.59)* | *3.99 (0.76)* | | *25.96 (1.73)* | *52.66 (12.08)** |

* indicates a significant difference between the Kanlow and Summer at the 0.05 probability level.
** indicates a significant difference between the Kanlow and Summer at the 0.01 probability level.
*** indicates a significant difference between the Kanlow and Summer at the 0.001 probability level.





Table 4. The $^{13}$C enrichment of aboveground plant biomass and belowground root biomass (ng
$^{13}$C g$^{-1}$ plant biomass) plus standard deviation (in parenthesis) for both switchgrass cultivars
Kanlow and Summer. P-values equal to or below 0.05 indicates significant difference between
cultivars within depth. DM = dry matter biomass (0% moisture).

|  |  | Kanlow | Summer |  |
|---|---|---|---|---|
|  |  | ng excess $^{13}$C g$^{-1}$ DM |  | P-value |
| Leaves |  | 474.43 (10.15) | 630.47 (113.19) | 0.069 |
| Tillers |  | 756.37 (110.11) | 1469.93 (252.99) | 0.007 |
| Crown |  | 4.69 (1.22) | 70.81 (39.38) | 0.003 |
| Roots | 0-10 | 9.96 (3.14) | 119.88 (54.09) | <0.0001 |
|  | 10-30 | 11.04 (1.65) | 76.56 (21.01) | 0.0002 |
|  | 30-60 | 16.21 (4.24) | 36.84 (8.82) | 0.0675 |
|  | 60-90 | 18.2 (11.04) | 29.12 (20.09) | 0.3544 |
|  | 90-120 | 8.66 (3.29) | 33.91 (34.34) | 0.0198 |
|  | 120-150 | 8.67 (2.48) | 26.24 (18.94) | 0.0907 |




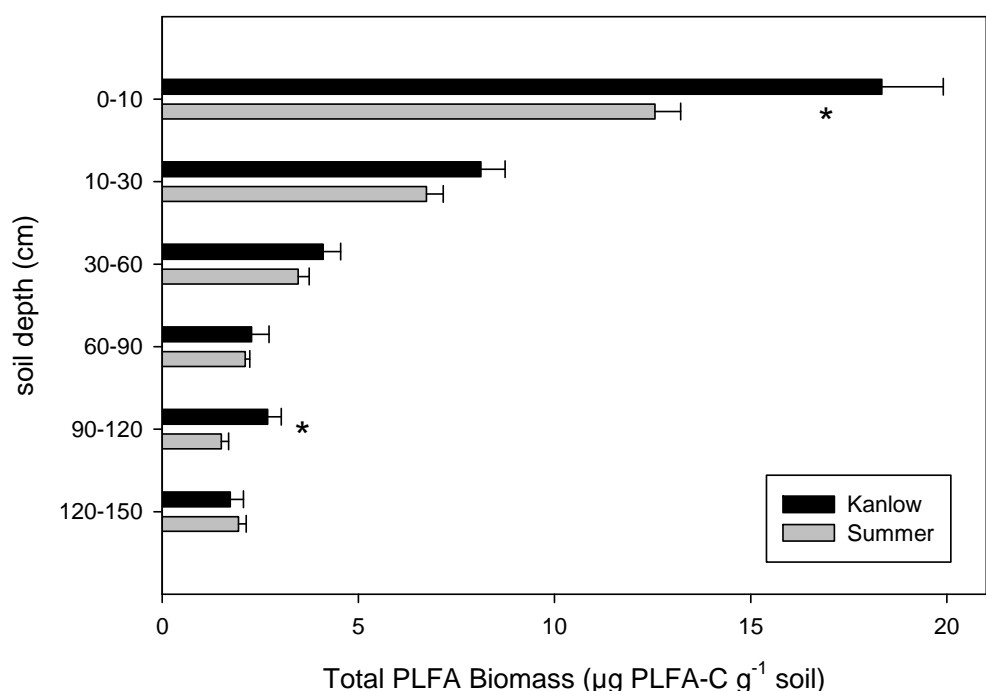

Figure 1. PLFA-derived C ($\mu$g PLFA-C g$^{-1}$ soil) for switchgrass cultivars Kanlow and Summer
by depth. Error bars represent standard deviations (n=3). * indicates a significance difference
between cultivars within depth.




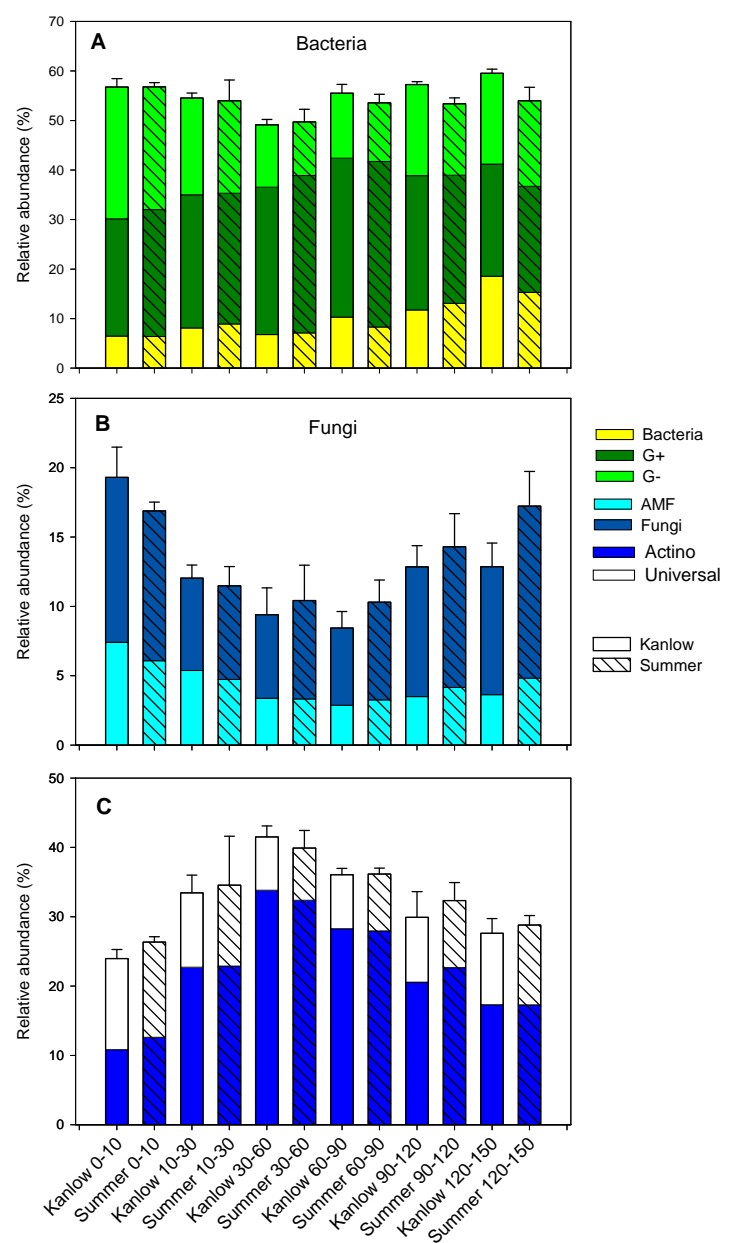

2 Figure 2. Soil microbial community composition (relative abundance, mol%) for switchgrass

3 cultivars Kanlow and Summer from 0-150 cm for A) bacterial groups, B) fungal groups and C)

4 actinomycetes and universal microbial groups. Error bars represent standard deviations (n=3).




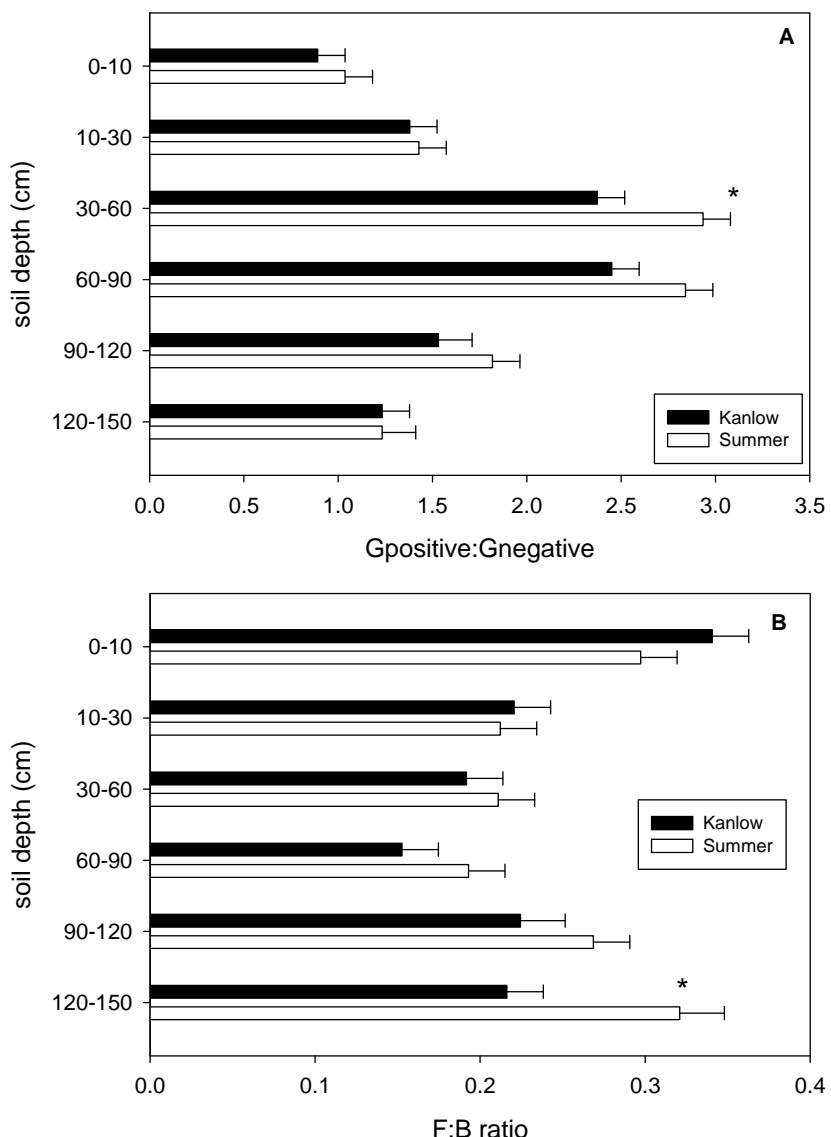

Figure 3. Gram-positive:gram-negative ratios (A) and fungal:bacterial ratios B) for switchgrass
cultivars Kanlow and Summer  by depth. * indicates a significant difference between cultivars
within depth.



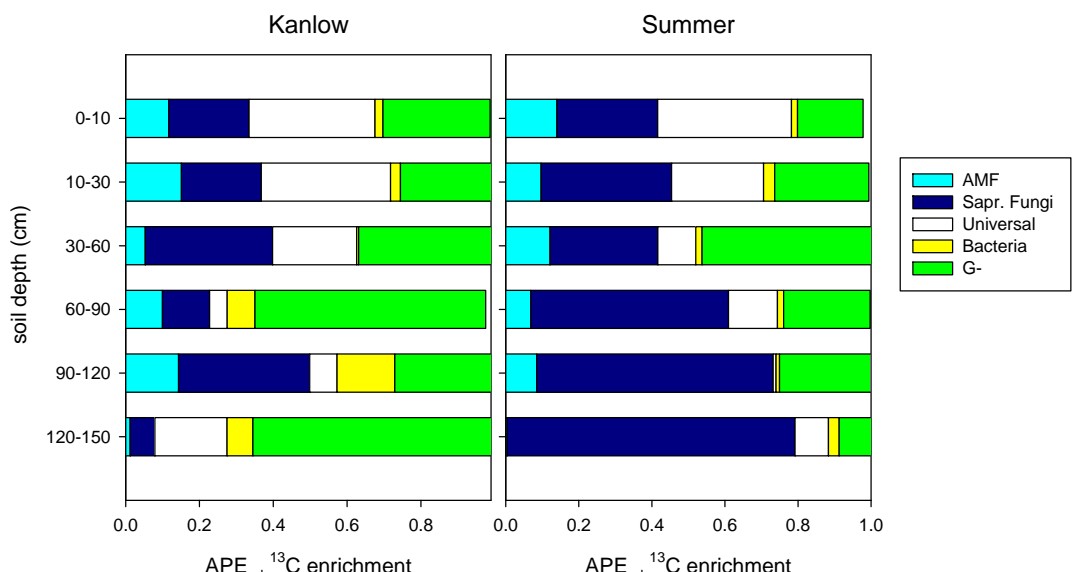

Figure 4. Relative rhizodeposit uptake (PLFA APE$_{rel}$ enrichment), for switchgrass cultivars
Kanlow and Summer at all sampled depths 48 hours after $^{13}$C labeling. Functional groups
actinomycetes and gram positive bacteria not included because $^{13}$C enrichment was not obtained
in those groups (Supplementary tables 1 and 2).