# Peer review of "Switchgrass ecotypes alter microbial contribution to deep"

_SOIL, 2015_

## Referee Comment (RC1) · Anonymous Referee #1 · 13 Feb 2016

General Comments This paper describes differences in switchgrass ecotypes with respect to biomass production, rooting structure, and soil microbial biomass and community structure plus its uptake of labeled exudates. The study is scientifically sound, using proper methods and suitable replication. A prodigious amount of work is behind the data.

Specific Comments 1. P2, Lines 8, 15. Substitute "biomass" for "abundance". PLFA is measured in ng of lipid biomarker per mass of soil and is commonly converted to biomass. It is not known to be directly related to cell abundances. For individual groups, it is measured as mole percent of the total and thus is a proportion of the total biomass. 2. P2, Lines 16-19. Please provide P values. 3. P2, Line 19. Insert "in" after "excess" 4. P2, Lines 29, 30 and P3 line 1. I think this statement is reversed and should be "greater productivity in lowland....". 5. Introduction. This study appears

to be examining a potential mechanism for varying soil carbon sequestration according to ecotype, i.e., root amount, structure, and exudates that may differentially affect soil microbial communities could alter the amount of soil C sequestered. What would be good to know at the onset is: What defensible data are there on the influence of switchgrass ecotypes (maybe even these same ones) on SOC at depths? 6. P3, lines 8, 12. There are many studies preceding Fierer et al (2003) that examine distributions of soil microbes with depth. A few examples are Federle et al., 1986, Wood et al., 1993, Dodds et al., 1996, Bone and Balkwill, 1988. Surely there are many in earlier decades. Maybe a review, a text, or earlier reference would be better for these general statements. 7. P4. Lines 13-17. Have these properties been previously measured in these specific ecotypes? This should be clarified in Introduction. 8. P4, line 29. Please clarify the timing of the burn with respect to the plant and soil sampling which follows. After May? 9. P8, line 1. No mention of neutral fractionation was made in the methods. 10. P10, line 6. Define SRL before use. 11. P10, lines 19-20. Please clarify that this was a transient increase in an otherwise downward trend. This statement is confusing after line 18. 12. P11, lines 1-4. The wording around these P values is objectionable to some readers who reject entirely "marginally significant". One option is to always state the P value and not provide an acceptable alpha in the methods and let the reader decide for themselves. A very sticky area; although, I personally have sufficient confidence in these effects. 13. P15, lines 9-16. Much of this text is verbatim from p13. Please re-write. 14. P15, line 18. Endophytes don't appear to be targeted by the sampling scheme. Re-write accordingly. 15. P15, lines 24-30. This paragraph switches between AMF and all fungi and is confusing as written. Please re-write. 16. P15, line 24. AMF biomarkers can be difficult to reliably use, especially PLFA 16:1w5cis. Please see Sharma and Buyer. Appl Soil Ecol. 2015. My recommendation is to downplay conclusions regarding AMF in this study. You have already shown that fungal biomarkers preferentially took up labeled exudates under Summer which is a nice finding and can be linked to C sequestration processes. 17. P16 lines 13-16. This info might be good in the Intro, especially if joined by other studies on ecotypes x SOC interaction. 18.

P16, line 24. Was aggregation measured in this study? Maybe insert "and therefore may promote..."

---

## Referee Comment (RC2) · Anonymous Referee #2 · 22 Feb 2016

This is a very interesting, novel and well-written paper. The significance of the study is fairly well articulated, with robust methods used to evaluate Switchgrass ecotype impacts on microbial communities. There are a few specific suggestions provided below that will help improved the manuscript. More could be made earlier on in the Abstract about the relevance of the research so that readers are drawn in. In places, particularly in the Introduction, the flow of text could be improved.

In Table 1 the carbon data should be presented on a volumetric basis. You should also determine the total carbon storage over the soil profile by accounting for bulk density. Express this on an area basis. If you have similar C contents over the profile after 3 years, yet vastly different root biomass, the result is highly significant to understanding how microbial decomposition versus root deposition of carbon affects carbon dynamics in soils.

[Figure]

Abstract Overall very well written and easy to follow.

Around line 5 the practical relevance of the altered microbial community would be useful to mention to broaden readership.

Line 11 - it is not known if biomass is per plant or per area. For this study I would argue that per soil area is of greater interest as plant density could vary. You probably state this later (I've not read the paper yet) but this should be clear in the Abstract.

Line 15 - Summer soils etc. - this is confusing to the reader as it implies a different soil rather than a soil planted with a different ecotype.

Introduction

This described the background to the research very well. Overall the flow is very good, but please review to see if you can make it a bit clearer. One suggestion is provided below. Hypotheses are clear.

page 4 - lines 2-3: You need to link these paragraphs more clearly. A bit of a jump at present.

Materials and Methods. Very well described. The experiment could be repeated with the information provided.

page 4, line 28 - include the planting density. page 8, line 1 - the fungi information is a Result and should be moved.

Results Again, well written and it describes the results well.

It would be easier to read if general categories of statistical significance were included: $P<0.001$, $P<0.01$, $P<0.05$, n.s.), e.g. page 9, line 20 - $P<0.001$ will suffice.

Discussion page 12, line 28 - change IL to 'Illinois in the US Midwest', so international readers can follow.

page 13, line 8 - there are plenty of studies on root traits versus soil properties so the

finding for lettuce was a bid odd to include when other studies exist for grasses grown in more similar conditions. From your data it appears that you can infer below-ground biomass from above-ground, so is yield not a simple measure to determine optimal ecotypes?

page 13, line 28 - change 'higher' to 'greater' to avoid confusing with depth. Check this throughout the paper as it appears in other places.

page 15, line 17 - you use 'cultivars' here and 'ecotypes' elsewhere. You are best to stick to one term, likely ecotype given the considerable phenotypic differences between the plant treatments.

page 15, line 29 and references - 'Rillig'

–––––––––––––––––––––––––––––

---

## Editor Comment (EC1) · P.D. Hallett (Editor) · 22 Feb 2016

Please see the 2 sets of referee comments provided for your manuscript. Both are favourable but suggest minor edits that would improve presentation. Please address these comments online and prepare a revised version of your paper. Best regards, Paul Hallett Technical Editor

---

## Author Comment (AC1) · 30 Mar 2016

Reviewer: RC1

Comments to the Author General Comments This paper describes differences in switchgrass ecotypes with respect to biomass production, rooting structure, and soil microbial biomass and community structure plus its uptake of labeled exudates. The study is scientifically sound, using proper methods and suitable replication. A prodigious amount of work is behind the data.

Specific Comments 1. P2, Lines 8, 15. Substitute "biomass" for "abundance". PLFA is measured in ng of lipid biomarker per mass of soil and is commonly converted to biomass. It is not known to be directly related to cell abundances. For individual groups, it is measured as mole percent of the total and thus is a proportion of the

total biomass.

Response: Change was made in the Abstract as suggested. See track changes, Page 2, Lines 10, 18.

2. P2, Lines 16-19. Please provide P values.

Response: We have added the p-value as requested. See track changes, Page 2, Lines 17-19.

3. P2, Line 19. Insert "in" after "excess"

Response: We have inserted the change as suggested. See track changes, Page 2, Line 22.

4. P2, Lines 29, 30 and P3 line 1. I think this statement is reversed and should be "greater productivity in lowland....".

Response: We have reversed the statement as suggested. See track changes, Page 3, Line 4.

5. Introduction. This study appears to be examining a potential mechanism for varying soil carbon sequestration according to ecotype, i.e., root amount, structure, and exudates that may differentially affect soil microbial communities could alter the amount of soil C sequestered. What would be good to know at the onset is: What defensible data are there on the influence of switchgrass ecotypes (maybe even these same ones) on SOC at depths?

Response: There is a surprising lack of data about cultivar effects on root biomass and SOC sequestration. Garten, who examined different lowland varieties, found no cultivar effect. We have added text to the introduction to emphasize the lack of data. See track changes, Page 3, Lines 8-14.

6. P3, lines 8, 12. There are many studies preceding Fierer et al (2003) that examine distributions of soil microbes with depth. A few examples are Federle et al., 1986,

Wood et al., 1993, Dodds et al., 1996, Bone and Balkwill, 1988. Surely there are many in earlier decades. Maybe a review, a text, or earlier reference would be better for these general statements.

Response: We have added in earlier references concerning the statements in question as suggested. See track changes, Page 3, Lines 16-17, 21.

7. P4. Lines 13-17. Have these properties been previously measured in these specific ecotypes? This should be clarified in Introduction.

Response: To our knowledge, this is the first use of stable-isotope probing to determine active microbial communities in-situ using these switchgrass cultivars. We have specified this per your suggestion. See track changes, Page 4, Lines 17-22.

8. P4, line 29. Please clarify the timing of the burn with respect to the plant and soil sampling which follows. After May?

Response: The plots were burned in early April. See track changes, Page 5, Line 13.

9. P8, line 1. No mention of neutral fractionation was made in the methods.

Response: See correction in track changes Page 7, Lines 13-14.

10. P10, line 6. Define SRL before use.

Response: See correction in track changes, Page 10, Line 19.

11. P10, lines 19-20. Please clarify that this was a transient increase in an otherwise downward trend. This statement is confusing after line 18.

Response: We have clarified the text as suggested. See track changes, Page 11, Lines 6-7.

12. P11, lines 1-4. The wording around these P values is objectionable to some readers who reject entirely "marginally significant". One option is to always state the P value and not provide an acceptable alpha in the methods and let the reader decide for themselves. A very sticky area; although, I personally have sufficient confidence in these effects.

Response: We have removed the word marginally, altered the method section, and kept the p-values. We understand that some reviewers may not accept the interpretation of the stats. See track changes.

13. P15, lines 9-16. Much of this text is verbatim from p13. Please re-write.

Response: This appears to be an editing error as a result of a previous re-write. We thank you for bringing it to our attention and have removed the text from Page 15. See track changes.

14. P15, line 18. Endophytes don't appear to be targeted by the sampling scheme. Re-write accordingly.

Response: The reviewer is correct that we did not sample endophytes. We have tempered the comment by adding "could have" to the text. See track changes, Page 16, Lines 7-9.

15. P15, lines 24-30. This paragraph switche between AMF and all fungi and is confusing as written. Please re-write.

Response: We have modified the text to emphasize the point that although we observed relatively modest differences between cultivars in fungal C uptake, these could be important in C cycling. See track changes, Page 16, Lines 14-21.

16. P15, line 24. AMF biomarkers can be difficult to reliably use, especially PLFA 16:1w5cis. Please see Sharma and Buyer. Appl Soil Ecol. 2015. My recommendation is to downplay conclusions regarding AMF in this study. You have already shown that fungal biomarkers preferentially took up labeled exudates under Summer which is a nice finding and can be linked to C sequestration processes.

Response: We have rewritten the paragraph to also include the saprotrophic fungi.

See track changes, Page 16, Lines 14-21.

17. P16 lines 13-16. This info might be good in the Intro, especially if joined by other studies on ecotypes x SOC interaction.

Response: We have edited the text as suggested however there is a surprising lack of data about cultivar effects and SOC sequestration. Garten, who examined different lowland varieties, found no cultivar effect. Ma et al. looked at SOC, but only one cultivar at two sites. See track changes, Page 3, Lines 8-14.

18. P16, line 24. Was aggregation measured in this study? Maybe insert "and therefore may promote..."

Response: No, unfortunately aggregation wasn't measured in this study. See track changes, Page 17, Line 22.

Reviewer: RC2

This is a very interesting, novel and well-written paper. The significance of the study is fairly well articulated, with robust methods used to evaluate Switchgrass ecotype impacts on microbial communities. There are a few specific suggestions provided below that will help improved the manuscript. More could be made earlier on in the Abstract about the relevance of the research so that readers are drawn in. In places, particularly in the Introduction, the flow of text could be improved.

In Table 1 the carbon data should be presented on a volumetric basis. You should also determine the total carbon storage over the soil profile by accounting for bulk density. Express this on an area basis. If you have similar C contents over the profile after 3 years, yet vastly different root biomass, the result is highly significant to understanding how microbial decomposition versus root deposition of carbon affects carbon dynamics in soils.

Response: Unfortunately, we do not have soil C data from the beginning of the experiment, so we cannot calculate a change over the 3 years. We agree that would be very useful data to have.

Abstract Overall very well written and easy to follow.

Around line 5 the practical relevance of the altered microbial community would be useful to mention to broaden readership.

Response: We added the practical relevance as suggested. See track changes, Page 2, lines 5-8.

Line 11 - it is not known if biomass is per plant or per area. For this study I would argue that per soil area is of greater interest as plant density could vary. You probably state this later (I've not read the paper yet) but this should be clear in the Abstract.

Response: We added units to the measurements referenced as suggested. See track changes. Page 2, Lines 14-15.

Line 15 - Summer soils etc. - this is confusing to the reader as it implies a different soil rather than a soil planted with a different ecotype.

Response: We have changed the phrasing to be clearer for the reader. See track changes, Page 2, Lines 17-19.

Introduction This described the background to the research very well. Overall the flow is very good, but please review to see if you can make it a bit clearer. One suggestion is provided below. Hypotheses are clear.

page 4 - lines 2-3: You need to link these paragraphs more clearly. A bit of a jump at present.

Response: We have made changes to the text to help the flow of the paper as suggested. See track changes, Page 4, Lines 11-12.

Materials and Methods. Very well described. The experiment could be repeated with the information provided.

page 4, line 28 - include the planting density.

Response: The plants were planted at a rate of 12 per 1m-2. See track changes, Page 5, Lines 10-11.

page 8, line 1 - the fungi information is a Result and should be moved.

Response: We see the reviewer's point of view, but choose to keep the NLFA sentence where it is in the method section. It is crucial to justifying our interpretation of the AMF data without adding yet more data to the manuscript.

Results Again, well written and it describes the results well.

It would be easier to read if general categories of statistical significance were included: P<0.001, P<0.01, P<0.05, n.s.), e.g. page 9, line 20 - P<0.001 will suffice.

Response: We believe that giving the actual p-value enables the reader to come to their own conclusion about the relative strength of the statistical relationships and have chosen not to group p-values as suggested.

Discussion page 12, line 28 - change IL to 'Illinois in the US Midwest', so international readers can follow.

Response: We have made this change as suggested. See track changes, Page 13, Line 17.

page 13, line 8 - there are plenty of studies on root traits versus soil properties so the finding for lettuce was a bid odd to include when other studies exist for grasses grown in more similar conditions. From your data it appears that you can infer below-ground biomass from above-ground, so is yield not a simple measure to determine optimal ecotypes?

Response: There are actually only a few studies that can document differences in root traits across different cultivars. We have removed the preceding sentence, which was confusing. The paragraph now focuses on cultivar effects. Yes, belowground biomass appears to be proportional to aboveground biomass. However, optimal ecotype determination may also take into account freeze-tolerance not just yield.

page 13, line 28 - change 'higher' to 'greater' to avoid confusing with depth. Check this throughout the paper as it appears in other places.

Response: We have made the suggested change here and throughout the manuscript. See track changes.

page 15, line 17 - you use 'cultivars' here and 'ecotypes' elsewhere. You are best to stick to one term, likely ecotype given the considerable phenotypic differences between the plant treatments.

Response: The terminology between ecotype and cultivar was confusing. We have modified the usage throughout the manuscript to specify ecotype in this manuscript (i.e. the difference between Summer and Kanlow), but cultivar when referring to other studies. This should clarify how our study differs from previous cultivar-specific studies. See track changes throughout.

page 15, line 29 and references - 'Rillig'

Response: We have made the change as requested. See track changes Page 16, Line 23 and References.

Reviewer: EC1

Please see the 2 sets of referee comments provided for your manuscript. Both are favourable but suggest minor edits that would improve presentation. Please address these comments online and prepare a revised version of your paper. Best regards, Paul Hallett Technical Editor

Response: Comments have been addressed and a revised manuscript provided.

Please also note the supplement to this comment:

http://www.soil-discuss.net/soil-2015-92/soil-2015-92-AC1-supplement.pdf

[Figure]

**Supplement:**

[revised manuscript text omitted]